# Evaluation of Clobetasol and Tacrolimus Treatments in an Imiquimod-Induced Psoriasis Rat Model

**DOI:** 10.3390/ijms25179254

**Published:** 2024-08-26

**Authors:** Philippe Guillaume, Tristan Rupp, Guillaume Froget, Sonia Goineau

**Affiliations:** Porsolt SAS, ZA de Glatigné, 53940 Le Genest-Saint-Isle, France; pguillaume@porsolt.com (P.G.); rupptristan@hotmail.fr (T.R.); gfroget@porsolt.com (G.F.)

**Keywords:** psoriasis, rat, preclinical model, Clobetasol, Tacrolimus, Imiquimod

## Abstract

Psoriasis is a chronic inflammatory skin disorder characterized by keratinocyte hyperproliferation, inflammation, and aberrant differentiation. Imiquimod-induced psoriasis in rodent models has been widely used to study the pathogenesis of the disease and evaluate potential therapeutic interventions. In this study, we investigated the efficacy of two commonly used treatments, Clobetasol and Tacrolimus, in ameliorating psoriatic symptoms in an Imiquimod-induced psoriasis Wistar rat model. Interestingly, rat models are poorly evaluated in the literature despite rats displaying several advantages in evaluating pharmacological substances. Psoriasis-like skin lesions were induced by topical application of Imiquimod cream on shaved dorsal skin for seven consecutive days. Following induction, rats in the treatment groups received either a Clobetasol or Tacrolimus ointment once daily for one week, while the control group did not receive any application. Disease severity was assessed using clinical scoring, histological examination, and measurement of proinflammatory cytokine levels. Both Clobetasol and Tacrolimus treatments significantly reduced psoriatic lesion severity compared to the control group. Clinical scoring revealed a decrease in erythema, scaling, transepidermal water loss, and thickness of skin lesions in both treatment groups with a more marked effect with Clobetasol. Histological analysis demonstrated reduced epidermal hyperplasia in treated animals compared to controls. Furthermore, Clobetasol led to a significant reduction in the expression levels of the interleukin-17 (IL-17a and IL-17f) proinflammatory cytokines in lesioned skin. Overall, our findings demonstrated the therapeutic efficacy of both Clobetasol and, in a modest manner, Tacrolimus in attenuating Imiquimod-induced psoriasis-like symptoms in a rat model. These results support the clinical use of these agents in the management of psoriasis and mitigating psoriatic inflammation. They also provide insights into the use of rats as a relevant species for the Imiquimod-induced psoriasis model.

## 1. Introduction

Psoriasis is an inflammatory skin disorder with an average prevalence of 1.5% in the US adult population and up to 1.92% in the European population [1,2]. Psoriasis’s etiology remains unclear, yet there is evidence of a strong genetic correlation with major histocompatibility complex and a number of environmental factors. Psoriasis-affected skin frequently exhibits enhanced angiogenesis, keratinocyte hyperproliferation in the epidermis, and immune cell infiltration in the dermis. Both innate and adaptive immunity are essential for the development of disease [3]. The functions of adaptive effector cells in psoriasis including regulatory T cells and Th17 cells are modulated by innate resident immune cells including neutrophils and dendritic cells. The inflammatory reactions in psoriasis are dependent on the IL-23/IL-17A inflammatory axis. Dendritic cells secrete IL-23 and activate IL-17-producing T cells. Notably, it was shown that γδ T cells are a primary producer of IL-17a, a cytokine that plays a significant role in skin disorders, inducing a cascade of cytokine expression associated with a pro-inflammatory signature through IL-17a and IL-17f secretions [3]. The immune cells are critical to induce psoriasis both in human and animal models.

The therapeutic arsenal is wide, but no curative treatment is currently available [4]. Failing this, treatments manage the symptoms of the disease by slowing the hyperproliferation of keratinocytes and reducing inflammation. Treatments can be divided into four classes: topical therapies, phototherapy, oral or parenteral systemic therapies, and biologics. Topical treatment is the first choice for mild to moderate psoriasis (<10% of body surface area affected) [5]. The side effects are mild (erythema, irritation, and burning of the skin) but the effectiveness is limited (no elimination of lesions in the long term). This causes patient dissatisfaction and poor compliance with treatments. Topical corticosteroids have an anti-inflammatory effect and inhibit the generation of pro-inflammatory cytokines at the transcriptomic level. They are the standard of care for mild to moderate psoriasis. Corticosteroids, including Clobetasol, along with macrolide calcineurin inhibitors, including Tacrolimus, are extensively used to treat psoriatic patients [4,6].

Spontaneous, genetically engineered human skin transplant and induced rodent psoriasis models inducing skin inflammation resembling psoriasis are being used to explore the mechanisms underlying the development of psoriasis and for testing potential new drugs [7]. A common in vivo preclinical model of psoriasis is the mouse model of psoriasis induced by Imiquimod (IMQ) [8]. IMQ, a Toll-like receptor 7/8 agonist, can induce or exacerbate psoriasis-like lesions mimicking many natural ligands that are involved in the induction of psoriasis. IMQ, usually administered topically, is used to create psoriatic-like lesions. This rapid and convenient model allows for the elucidation of underlying mechanisms and the evaluation of new therapies against psoriasis. The psoriasis area severity index (PASI) score, as well as ear or skin thickness measurements and histological staining, have been very well established in this murine model [6,9].

Most of the studies focus on mouse models of psoriasis while rat models are poorly investigated [7,8]. There are a limited number of articles that investigate models of psoriasis-like structures in the rat [9,10]. To our knowledge, no study has evaluated the effect of Clobetasol and Tacrolimus in an IMQ-induced psoriasis model in rats. In this work, our data showed that both treatments decreased the psoriatic lesion severity induced by topical application of IMQ when compared to the IMQ-treated group. Clinical assessment indicated a notable reduction in erythema, scaling, transepidermal water loss, and skin lesion thickness in both treatment cohorts. Additionally, histological examination revealed a decrease in epidermal hyperplasia in treated animals in contrast to the IMQ-treated group. Moreover, Clobetasol treatment was associated with a substantial and either transient or constant decline in the expression levels of proinflammatory cytokines in the affected skin, including IL-17a and IL-17f.

## 2. Results

### 2.1. Evaluation of Imiquimod Effect on Rat Body Weight and Scratching Behavior upon Anti-Psoriatic Drug, Clobetasol, and Tacrolimus

We demonstrated that IMQ induced a reduction in body gain which is not significant for female rats and significant for male rats from Day 3 (Figure 1A and Appendix A). We also observed that Clobetasol treatment aggravated this body weight loss compared to IMQ alone for both genders (Figure 1A and Appendix A). We also evaluated scratching behavior in rats, given that pruritus is one of the most common and distressing symptoms of psoriasis in humans, significantly affecting the quality of life. The scratching evaluation directly assesses this symptom, providing a relevant measure of disease impact [11]. We demonstrated that scratching behavior, evaluated on Day 4, was significantly increased in the IMQ-treated group, with the total number of bouts of scratching reaching 120.0 ± 26.9 versus 16.1 ± 4.4 in the control group (Figure 1B). On Day 7, scratching behavior was still increased in the IMQ-treated group (83.8 ± 19.7 versus 16.0 ± 2.7 in bouts of scratching the control group), while lower than that observed on Day 4, and did not reach statistical significance compared to the control group (*p* = 0.0872) (Figure 1B). Clobetasol reduced IMQ-induced scratching behavior evaluated on Day 4, the number of the bouts of scratching reaching 49.0 ± 13.6 versus 120.0 ± 26.9 in the IMQ-treated group (i.e., −59%). The level of scratching measured on Day 7 remained similar (49.0 ± 13.6 bouts of scratching); however, the reduction did not reach statistical significance, presumably because scratching behavior was lowered from Day 4 to Day 7 in the IMQ-treated group. Tacrolimus tended to reduce IMQ-induced scratching behavior evaluated on Day 4, with the number of the bouts of scratching reaching 72.0 ± 23.1 versus 120.0 ± 26.9 in the IMQ-treated group (i.e., −40%, NS). In contrast, no reduction was observed on Day 7 with Tacrolimus (Figure 1B). Conversely, in male rats, we also observed an increase in scratching behavior following the IMQ application that did not reach significance (Appendix A). Interestingly, the scratching was significantly promoted by Clobetasol, which is not consistent with the results observed in female rats (Appendix A).

### 2.2. Evaluation of Psoriasis-like Inflammation Severity Index (PASI)

The PASI score provides a comprehensive measure of both the severity and the extent of psoriasis and is used in both human and animal models as a translational tool to evaluate psoriasis severity [7,12]. It assesses the degree of erythema (redness), induration (thickness), and desquamation (scaling) in different body regions, offering a holistic view of the disease. PASI is a standardized tool used globally, which ensures consistency and comparability across clinical studies, allowing for reliable assessment of disease severity and treatment efficacy [13]. IMQ induced erythema, scaling, and thickening of the skin, with maximum effects reached on Days 3 and 5, depending on the parameters (Figure 2A and Appendix A). The maximum scores reached 3.0 ± 0.0 for erythema, 3.4 ± 0.3 for scaling, and 3.0 ± 0.2 for thickening (Figure 2B–D). The mean PASI score increased progressively to reach a maximum on Day 5 (9.4 ± 0.4). Thereafter, despite the continuous application of IMQ, PASI was slightly reduced to reach 7.3 ± 0.6 on Day 7 (Figure 2A). In the control group, no erythema, thickening, or scaling of the skin was observed over the test period (Appendix A). In addition, the mean PASI score was null (Figure 2A). Both Clobetasol and Tacrolimus significantly reduced PASI score compared to IMQ alone even if Clobetasol displayed a significantly stronger effect (Figure 2A). Clobetasol significantly reduced from Day 3, erythema (0.9 ± 0.2 versus 3.0 ± 0.0 in the IMQ-treated group at Day 3, *p* < 0.001), scaling (0.9 ± 0.1 versus 3.4 ± 0.3 in the IMQ-treated group at Day 5, *p* < 0.001) and thickening (0.4 ± 0.2 versus 3.0 ± 0.2 in the IMQ-treated group, *p* < 0.001). The mean PASI was reduced from Day 3 with a maximum effect on Day 5 (2.4 ± 0.3 versus 9.4 ± 0.4 in the IMQ-treated group, *p* < 0.001). The beneficial effects of Clobetasol were maintained up to the end of the testing period (Day 7) (Figure 2 and Appendix A). The same effects occurred upon IMQ also in males in a significant manner (Appendix A). Nevertheless, the effect of Clobetasol is more limited, even if significant, compared to what was observed in female rats, mainly due to the effect on thickening and limited effect, if any, on scaling and erythema (Figure 2C,D and Appendix A). Tacrolimus significantly reduced thickening from Day 5 (1.8 ± 0.3 versus 2.8 ± 0.2 in the IMQ-treated group on Day 7), scaling from Day 3 (0.9 ± 0.4 versus 3.4 ± 0.3 in the IMQ-treated group on Day 5), and erythema on Day 7 (0.5 ± 0.3 versus 2.4 ± 0.2 in the IMQ-treated group). The mean PASI was significantly reduced from Day 5 with a maximum effect on Day 7 (2.8 ± 0.8 versus 7.3 ± 0.6 in the IMQ-treated group) (Figure 2). The beneficial effects of the Tacrolimus treatment were maintained up to the end of the testing period (Day 7), although it was less effective than Clobetasol. The effects occurred earlier in females compared to males (Appendix A).

### 2.3. Evaluation of Skin Thickness, Erythema, and Transepidermal Water Loss (TEWL)

To better characterize the impact of skin inflammation, we assessed various parameters including skin thickness, erythema, and water loss, which are useful tools to determine disease severity [11]. The skin thickness progressively increased in the IMQ-treated group, with maximum effect reached on Day 3 (Figure 3A). Thereafter, the thickness was very slightly reduced. According to the PASI score, the maximum skin thickness was observed on Day 5. Clobetasol significantly reduced the skin thickness from Day 3 compared to the IMQ-treated group and the effect was maintained up to the end of the testing period (Day 7). In contrast, Tacrolimus had no effect on skin thickness. The skin erythema, quantified as an index using the Mexameter, progressively increased up to Day 5 in the IMQ-treated group. Thereafter, the erythema index was slightly reduced (Figure 3B). These findings are in line with the erythema score observed when using the PASI scoring. Both Clobetasol and Tacrolimus significantly reduced the skin erythema index compared to the IMQ-treated group. The beneficial effect of both treatments was quite similar (−24% maximum on Day 3 for Clobetasol; −20% maximum on Day 5 for Tacrolimus) and was maintained up to the end of the testing period (Day 7). TEWL progressively increased in the IMQ-treated group, with the maximum effect reached on Day 3 (120.3 ± 25.4 versus 17.6 ± 1.9 g/m^2^·h in the control group) (Figure 3). Thereafter, despite IMQ still being applied, TEWL progressively reduced. Both Clobetasol and Tacrolimus significantly reduced TEWL compared to the IMQ-treated group. The beneficial effect was slightly higher with Tacrolimus (−61% maximum on Day 3 for Clobetasol; −80% maximum on Day 3 for Tacrolimus) and was maintained up to the end of the testing period (Day 7) with both treatments.

### 2.4. Evaluation of Epidermal Thickness (Histology)

We performed a histological analysis using Hematoxylin and Eosin (HE) staining upon treatments to determine the roles of Clobetasol and Tacrolimus in the psoriasis-like skin pathology and IMQ-induced thickening of the epidermis (Figure 4A). With this method, we evaluated epidermal thickness on Day 7 and observed a significant increase in the IMQ-treated group (58.9 ± 2.1 versus 23.8 ± 1.2 µm in the control group, i.e., +147%) (Figure 4B). Interestingly, both Clobetasol and Tacrolimus significantly reduced IMQ-induced epidermal thickening. The beneficial effect was slightly higher with Clobetasol (34.0 ± 2.0 µm) than with Tacrolimus (49.7 ± 3.5 µm) compared to the IMQ-treated group (58.9 ± 2.1 µm), i.e., −42% and −16%, respectively).

### 2.5. Cytokine Profile

In order to better understand the mechanisms by which Clobetasol- and Tacrolimus-mediated signaling repress psoriasis in our model, we examined the mRNA expression of IL-17a and IL-17f that are known potent cytokines in psoriatic plaques [14]. The IMQ treatment led to significant increases in the expression of both IL-17a and IL-17f compared to control rats on Day 7 (Figure 5A,B). We observed this effect from Day 3, which reached statistical significance on Day 5 (Appendix A). Clobetasol abolished IMQ-dependent IL-17a and IL-17f upregulation (Figure 5A,B and Appendix A). In contrast, Tacrolimus treatment did not influence IL-17a and IL-17f upregulation upon IMQ, indicating that IL-17 pathways are not involved in the moderate anti-psoriatic effect observed (Figure 5A,B). These findings suggest that Clobetasol interfered with IL-17 pathways and drove the anti-psoriatic effect.

## 3. Discussion

Psoriasis is primarily characterized by the thickening of the epidermis, along with the infiltration of inflammatory cells into the dermis and epidermis leading to the formation of red scaly plaques [12]. Psoriasis is marked by an increase in epidermal cell proliferation and the infiltration of immune cells within the dermis. The underlying mechanisms of psoriasis are intricate, involving a complex interaction among keratinocytes, immune cells, and various resident skin cells [15]. Recent studies have shown that psoriasis has been recognized primarily as a disease driven by immune cells, with keratinocytes acting merely as facilitators of immune functions involving the IL-23/IL-17 signaling axis [14,16]. The activation of plasmacytoid dendritic cells enhances the maturation of myeloid dendritic cells and the subsequent production of pro-inflammatory cytokines. This cascade activates T helper cells, specifically Th1 and Th17, inducing the secretion of additional inflammatory cytokines [14]. These cytokines, particularly IL-17, activate keratinocytes that secrete antimicrobial peptides, cytokines, and chemokines that further exacerbate inflammation [17]. Preclinical models mimicking human psoriasis have been developed and utilized over the years. Topical application of IMQ on mouse skin leads to the rapid proliferation of plasmacytoid dendritic cells and stimulates keratinocytes to increase cytokine production [18,19,20]. These effects in the rodent skin closely resemble human plaque-type psoriasis with respect to erythema, skin thickening, scaling, and epidermal alteration. Most of the current work has been conducted on mouse models of psoriasis; however, the effect on rats has been, so far, poorly investigated with limited pharmacological data [10,11,16,21,22]. In our study, IMQ was used as an inducer of psoriasis-like symptoms and evaluated in Wistar rats. This study demonstrated that the topical application of IMQ for seven consecutive days resulted in skin changes that mimic human psoriasis, including thickening of the epidermal layer, erythema, scaling of the skin, redness, or water loss. The histopathological investigation showed epidermis hyperplasia and hypertrophy in the dorsal rat skin, as previously reported in the mouse [23,24]. It is to be noted that the effects tended to decline after five days of IMQ treatment, suggesting an adaptive reaction of the skin to IMQ stimulation, as already described in the mouse model [25]. This local physical phenomenon was accompanied by pro-inflammatory cytokine upregulation (IL-17a and IL-17f). Additionally, there was a significant increase in the PASI score used in the clinic [26].

The PASI score, ranging from 0 to 72, is a widely used tool in clinical dermatology to evaluate the severity of psoriasis, combining the assessment of the severity and extent of the lesions [4]. This score, which is adapted by modifying the assessment criteria to match the anatomical and physiological differences between humans and animals, is also used in preclinical studies to monitor the progression or regression of the disease. The PASI score should provide a standardized method to quantify and compare the severity of psoriasis in both clinical and preclinical studies. However, the PASI-like scoring system used in preclinical assays can be subjected to various subjective biases related to the researcher potentially affecting the reliability and validity of the results. For these reasons, we also quantified the skin thickness, erythema, and the loss of water using a quantitative method. Both approaches showed a similar profile with an increase in skin erythema and thickness. However, the kinetics of the effects seem different, with a maximum effect occurring earlier with the quantitative method. Differences in kinetics between both methods have already been reported [27]. Although the PASI score remains the scoring system of choice in the psoriasis preclinical model, the addition of quantitative determination of the lesions can confirm the skin damage in a more accurate way.

Most of the preclinical studies investigating human psoriasis were conducted in mice, including spontaneous models, genetically engineered mice, or xenograft models [7]. Nevertheless, the work remains limited to the rat species [7]. The model of daily application of Aldara cream (IMQ) to the depilated backs of animals has several advantages, the first one being that it is cheap, very easy to use, and rapidly develops the pathology. Our choice to select the Wistar rat was driven by several scientific and practical considerations. Rats offer many advantages over mice as they are larger than mice, providing more skin surface area for the application of IMQ and making it easier to perform repeated applications, biopsies, and other manipulations. The more extensive blood sampling capacity of rats has, on several occasions, allowed for the following of the pharmacokinetic or inflammatory profiles for longitudinal studies. The cytokine response in rats also closely mimics that observed in humans, particularly in the context of Th17-mediated inflammation involved in psoriasis. Although we tested both genders, female rats were selected for the pharmacological studies, as the hair regrowth was less marked in the females compared to the males and the skin inflammation occurred earlier in females. In addition, in the male rats, IMQ failed to induce scratching behavior.

Using our rat model, this study also investigated the psoriasis of the responses of two standards of care, Clobetasol and Tacrolimus. Both treatments decreased the psoriatic lesion severity induced by topical application of IMQ when compared to the control group. Scratching behavior and PASI score, including skin thickness, scaling, and erythema, were reduced, and the thinning of the epidermis was also confirmed through histopathology in both treatment cohorts. However, the effects were more marked in the Clobetasol-treated group compared to the Tacrolimus-treated group. In contrast, the expression of pro-inflammatory cytokines (IL-17a and IL-17f gene expression) was only decreased in the Clobetasol-treated group from Day 5. Indeed, Tacrolimus had no effects on the levels of IL-17a and IL-17f on Day 7. The decrease in IMQ-induced psoriasis-like inflammation by Clobetasol was in accordance with published data using mice [28,29,30]. Clobetasol has played a central role over the past several decades as a treatment for many dermatologic disorders [31]. It is a potent topical corticosteroid therapy commonly used to treat various inflammatory skin conditions like psoriatic lesions in humans [32,33], involving several pathways that contribute to its anti-inflammatory, immunosuppressive, and antiproliferative effects. We also observed that Clobetasol enhanced body weight loss upon IMQ, suggesting Clobetasol-mediated toxicity as observed in a mouse model of psoriasis [34]. Indeed, Clobetasol is a potent topical corticosteroid, and the ingestion of corticosteroids by the animal may have a toxic metabolic effect, reducing both food intake and affecting glucogenesis, leading to body weight loss [35]. The effects of Clobetasol exhibit comparable effects on body weight and PASI score between males and females; nevertheless, differences in scratching behavior can be observed (Figure 1, Figure 2, Appendix A). Indeed, males tend to experience increased scratching behavior. In previous work, we already observed a decrease in scratching behavior upon Clobetasol in female rats [11]. We think that this absence of expected effect might be due to hypersensitivity in male rats; indeed, psoriasis induces mild chronic stress due to the pathogenesis and associated disorders [36]. Interestingly, male rats are more sensitive to stress than female rats, which can worsen the symptoms, including itch hypersensitivity [37]. Thus, we think that, despite a consistent global response between males and females upon Clobetasol, the effect on scratching might be delayed, and further analysis using longer treatment might be required to observe an effect on scratching behavior in male rats. 

In our study, Tacrolimus showed moderate effects on the psoriatic lesions, in accordance with published data in the mouse [38]. Tacrolimus, a macrolide immunosuppressive inhibitor of calcineurin, arises as an alternative for the treatment of psoriasis, acting on some cytokines involved in the pathogenesis of the disease. It inhibits the activation of T lymphocytes and their derived Th2-related cytokines and Th1-related cytokines [39]. More potent and with fewer secondary effects than cyclosporine, it has been the object of considerable interest. Although Tacrolimus has been shown to be an effective monotherapy for psoriasis, its performance can vary and may not always give satisfactory results in some patients [40]. Currently, combination therapy is frequently used to manage psoriasis because clinical trials have shown that it may provide synergistic benefits and reduce risks of adverse effects [40]. In our study, Tacrolimus showed a poor effect on the skin’s IL-17 levels, whereas IL-17 cytokines are known to have a crucial role in psoriasis pathogenesis. Indeed, its skin expression is closely related to the development of skin lesions [23]. The lack of effects of Tacrolimus on IL-17 levels could explain the poor effects measured on the psoriatic lesions in our study (moderate effects on skin thickening or erythema). Tacrolimus is known as a challenging molecule due to low solubility, low penetration, poor bioavailability, and toxicity [41]. In humans, the efficacy of topical Tacrolimus ointment for chronic psoriasis plaques is controversial because the absorption of this drug is hindered by the thick scales of psoriasis [42,43]. This could explain, in part, the poor effect of Tacrolimus in our study, which was topically applied approximately 1 h after IMQ application. 

Fortunately, the evolving landscape of psoriasis treatment presents significant potential for enhanced patient outcomes. Recent developments include the introduction of innovative biologic agents that target new pathways, the investigation of combination therapies aimed at increasing effectiveness while reducing adverse effects, the application of biomarkers for guiding treatment choices and monitoring progress, and the progression of gene- and cell-based therapeutic approaches [5]. In particular, innovative biological agents that focus on novel pathways, including Mirikizumab, an interleukin 23 inhibitor, or RORγt and ROCK2 inhibitors demonstrate potent effectiveness and might serve in the near future as therapeutic alternatives [4].

## 4. Materials and Methods

### 4.1. Animals

Female Wistar (Han) rats of 3 months old (203–296 g) and male Wistar (Han) rats of 3 months old (232–256 g) were supplied by Janvier Labs (Le Genest-Saint-Isle, France). Animals were acclimated at least 5 days before the experiments. The animals were housed in cages containing wood litter (Safe, Augy, France) with free access to food (Code A04—Safe, Augy, France) and tap water. Nesting enrichment was provided, including a tunnel, gnawing material, and nesting material. The animal room was maintained under artificial lighting between 7:00 and 19:00 at a controlled ambient temperature of 22 ± 2 °C and relative humidity between 20 and 80%. The number of animals per group is included in the figure legends for all the experimental designs.

### 4.2. Animal Ethical Consideration and Limit Points

Experiments were conducted in strict accordance with Council Directive No. 2010/63/UE of 22 September 2010, and the French decree No. 2013-118 of 1 February 2013, on the protection of animals for use and care of laboratory animals and their scientific use. The study was performed in a lab accredited by the Association for Assessment and Accreditation of Laboratory Animal Care (AAALAC). All experiments were also approved by the ethics committee for animal experimentation (agreement n° F 53 1031). The following parameters were considered as limit points that required rat sacrifice by CO_2_ inhalation or cervical dislocation: a body weight loss greater than 20% relative to the initial weight, ulceration, hypothermia (<34 °C), or failure to eat and drink.

### 4.3. Psoriasis Induction

One or 2 days before the start of topical application of IMQ, the rats were anesthetized (isoflurane) and had their central part of the back shaved and depilated using a depilatory cream (on an area of approximately 4 cm^2^). Once shaved, rats were housed singly until the end of the experiment. Rats then received a daily topical dose of 37.5 mg/cm^2^ of 5% Aldara^®^ IMQ cream (Meda Pharma, Mérignac, France) on the shaved back (i.e., 150 mg of IMQ cream onto the area of 4 cm^2^ dilapidated skin) for up to 7 days, depending on the experiments. A control group of rats did not receive IMQ.

### 4.4. Study Design

Female rats received IMQ for up to 7 consecutive days from Day 0 to Day 6, and the animals were divided into 3 groups (groups 2 to 4; n = 8 per group). A control group was included for comparison (n = 8).

The treatment groups were, therefore, as follows (first experiment):-Group 1 (Control group): no IMQ + no treatment-Group 2 (IMQ-treated group): IMQ + no treatment-Group 3 (Clobetasol group): IMQ + Clobetasol propionate-Group 4 (Tacrolimus group): IMQ + Tacrolimus monohydrate

The control female rat did not receive any application on the skin. In group 3, 120 mg of 0.05% Dermoval^®^ cream (Clobetasol propionate, GSK France reference n° 3400932043354, Rueil-Malmaison, France) was applied topically on the back area (i.e., 30 mg/cm^2^), approximately 1 h after IMQ application and once daily for 7 consecutive days from Day 0 to Day 6. In group 4, 160 mg of 0.1% Protopic^®^ ointment (Tacrolimus monohydrate, Leo Pharma reference n° 3400935922311, Ballerup, Denmark) were applied topically on the back area (i.e., 40 mg/cm^2^), approximately 1 h after IMQ application and once daily. In a second experiment, a second batch of female rats (n = 6 per group) was prepared to better characterize the kinetic profile of the cytokines. Groups 1 to 3 (with the same treatments as in the first experiment) were evaluated again, but the duration of applications was either 3 or 5 days (sacrifice of the animals at different time points). In a separate study, male rats were evaluated under similar experimental conditions, but only the effects of Clobetasol were investigated in comparison to the control or IMQ-treated group (n = 8 rats per group). Control male rats received Vaseline application on the skin (Figure 6).

### 4.5. Evaluation of Psoriasis-like Inflammation Severity Index (PASI)

To score the severity of the inflammation of the back skin, a scoring system, based on the clinical Psoriasis Area and Severity Index (PASI), was used. Erythema, scaling, and thickening were scored independently from 0 to 4 (0: none; 1: slight; 2: moderate; 3: marked; 4: very marked). The cumulative score (erythema plus scaling plus thickening) served to indicate the severity of inflammation (scale 0–12). Scoring was performed prior to the daily treatment application on Day 0, Day 2, Day 3, Day 5, and Day 7.

### 4.6. Evaluation of Skin Erythema and Transepidermal Water Loss (TEWL)

In addition to the erythema evaluation in the context of the PASI score, skin erythema was also quantified in isoflurane-anesthetized rats using a modular skin testing system (Mexameter MX18 MDD, Monaderm, Monaco). Transepidermal water loss (TEWL) was measured using a TEWL device (Vapometer, Delfin Technologies Ltd., Kuopio, Finland). Each measurement was carried out in triplicate (mean values are reported). TEWL and skin erythema were measured on similar days as those for the PASI evaluation.

### 4.7. Evaluation of Scratching Behavior

Scratching behavior was evaluated on Day 4 and Day 7. On each testing day, rats were placed individually in an observation chamber (cylinder: height = 35 cm, diameter = 19 cm) for at least 60 min to habituate to the testing environment. They were then videotaped for 120 min to assess scratching. Technicians were not present in the observation room during the videotaping. A bout of scratching was defined as one or more rapid movements of the hind paws directed toward and contacting the treated area (for any observable length of time), ending with licking, or biting of the toes and/or placement of the hind paw on the floor. Hind paw movements directed away from the site (ear-scratching) and grooming movements were not counted. The degree of scratching was quantified as the total number of bouts of scratching over the 120-min observation period.

### 4.8. Evaluation of Epidermal Thickness by Histology

At the end of the in vivo evaluation period, the rats were sacrificed, and a skin sample was collected. The segment was fixed in 10% neutral buffered formalin for 48 h. The samples were rinsed 3 times in 0.1 M phosphate-buffered saline (PBS, pH 7.4), stored in 0.1 M PBS at 4 °C, and embedded in paraffin. Tissue sections of 4 µm were prepared and stained with Hematoxylin and Eosin (HE). Whole stained sections were digitally acquired using the VS200 slide scanner (Olympus France, Rungis, France) at a 200× magnification. Epidermal thickness was measured (mean value of three areas per section) using image analysis (proprietary algorithm).

### 4.9. Quantification of Inflammatory Cytokine in Skin Samples by qPCR

A piece of skin sample was collected from each animal and precisely weighed on Day 3, Day 5, and Day 7. The tissue pieces were rinsed with cold PBS and incubated in RNAlater reagent at 4 °C for a maximum of 24 h. The tissue was mechanically homogenized in a Lysing Matrix D tube in cold RLT lysis buffer (116913050/116933005, MP Biomedicals France, Illkirch, France) using the FastPrep-24™ 5G system (MP Biomedicals France, Illkirch, France). Supernatants were added onto a QIAshredder column (79656, Qiagen France, Courtaboeuf, France) before total RNA purification using the RNeasy Plus Mini kit (74136, QIAGEN) following the supplier’s recommendations. Purified RNA was quantified using Nanodrop™ 2000c device. Total RNA was reverse transcribed using Maxima H Minus First Strand cDNA Synthesis Kit (K1682, Thermo Fisher Scientific France, Saint Herblain, France) as per the supplier’s recommendations. IL-17a and IL-17f expression were quantified using the TaqMan Fast Advanced Master Mix (557, Thermo Fisher Scientific France, Saint Herblain, France) and target-specific primers and probe (Table 1) in duplicate. Amplification was performed using QuantStudio™ 5 Real-Time PCR System (Thermo Fisher Scientific France, Saint Herblain, France). The method to evaluate gene expression was adapted from Boulter et al. [44]. Copy numbers for both targets of interest were extrapolated using a standard curve (gBlocks™ Gene Fragments from the target genes, Integrated DNA Technologies, Coralville, IA, USA) and normalized to the copy number of GAPDH in the same sample (multiplex reactions).

### 4.10. Statistical Analysis

Statistical analyses and graphical representations were performed using GraphPad Prism (version 10.1.1 or higher, GraphPad Software Inc., Boston, MA, USA). *p*-values < 0.05 were considered as statistically significant (* *p* < 0.05; ** *p* < 0.01; *** *p* < 0.001; **** *p* < 0.0001). Data were tested for normality using the D’Agostino–Pearson test prior to the use of parametric tests. If normality was not confirmed, a non-parametric test was used. Body weight, scratching, PASI score, skin thickness, skin erythema, and TEWL were analyzed for each day with the mixed-effect model using restricted or residual maximum likelihood (REML), including group and day as factors with repeated measures alongside Tukey’s multiple comparison tests, for each day. For epidermal thickness, data were analyzed using a one-way ANOVA, including the groups as factors, and intergroup differences were evaluated with Tukey’s multiple comparison tests. For qPCR, data were analyzed using a Kruskal–Wallis test, and intergroup difference was evaluated with Dunn’s multiple comparison tests on Day 7 or with the mixed-effect model using REML including group and day as factors and followed by Tukey’s multiple comparison tests for each day.

## 5. Conclusions

In conclusion, the present study demonstrated the utility of the rat as a relevant species for the Imiquimod-induced psoriasis model. This rapid and convenient model can help decipher the underlying mechanisms and evaluate new therapies against psoriasis. The therapeutic efficacy of both Clobetasol and, in a modest manner, Tacrolimus in attenuating Imiquimod-induced psoriasis-like symptoms has been shown in this rat model. This comprehensive preclinical model is therefore suitable as an initial in vivo screening model for different anti-psoriasis pharmaceutical drugs. This work provides new insights into the biological effects of the IMQ-induced, psoriasis-like phenotype and its reversal using Clobetasol treatment in Wistar rats. We also demonstrated that Tacrolimus had a very limited anti-psoriatic effect. We trust that these encouraging data will stimulate the psoriasis research community to further evaluate novel anti-psoriasis drugs.

## Figures and Tables

**Figure 1 ijms-25-09254-f001:**
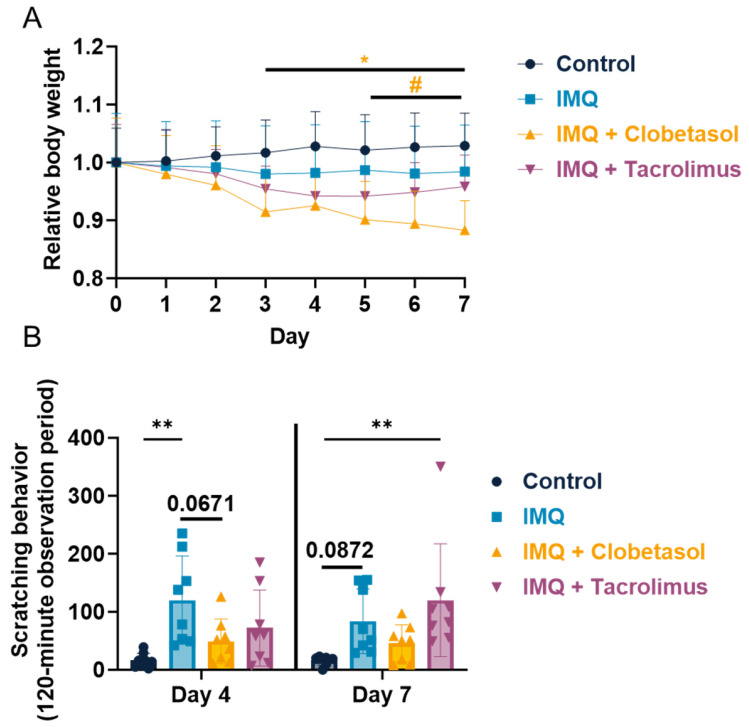
Effects of treatments on female rats’ body weight and scratching behavior upon Imiquimod: (**A**) Evaluation of rat body weight upon Imiquimod (IMQ, topical administration, 37.5 mg/cm^2^) with or without Clobetasol (topical administration, 30 mg/cm^2^) or Tacrolimus (topical administration, 40 mg/cm^2^). Treatments were administered daily for up to 7 days. (**B**) Evaluation of scratching behavior upon treatments. Statistical differences between the groups were determined using a mixed-effects model (REML, groups, and time as a factor) followed by Tukey’s multiple comparisons test: * *p* ≤ 0.05 control vs. IMQ + Clobetasol, # *p* ≤ 0.05 IMQ vs. IMQ + Clobetasol, and ** *p* ≤ 0.01. Data represent mean and SD. n = 8 rats per group.

**Figure 2 ijms-25-09254-f002:**
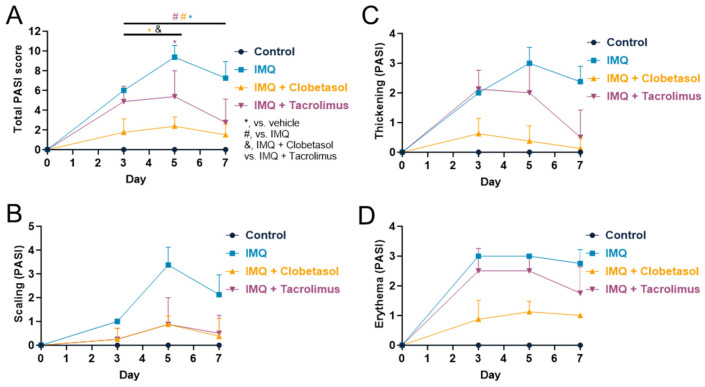
Effects of treatments on female rats’ PASI score upon Imiquimod: (**A**) Evaluation of the cumulative total PASI score (including thickening, scaling, and erythema) upon Imiquimod (IMQ, topical administration, 37.5 mg/cm^2^) with or without Clobetasol (topical administration, 30 mg/cm^2^) or Tacrolimus (topical administration, 40 mg/cm^2^). Treatments were administered daily for up to 7 days. (**B**) Evaluation of thickening upon treatments. (**C**) Evaluation of scaling upon treatments. (**D**) Evaluation of erythema upon treatments. Statistical differences between the groups were determined using a mixed-effects model (REML, groups, and time as a factor) followed by Tukey’s multiple comparisons test: * *p* ≤ 0.05 control vs. treatments, # *p* ≤ 0.05 IMQ vs. IMQ + Clobetasol or IMQ + Tacrolimus, and & *p* ≤ 0.05 IMQ + Clobetasol vs. IMQ + Tacrolimus. Data represent mean and SD. n = 8 rats per group.

**Figure 3 ijms-25-09254-f003:**
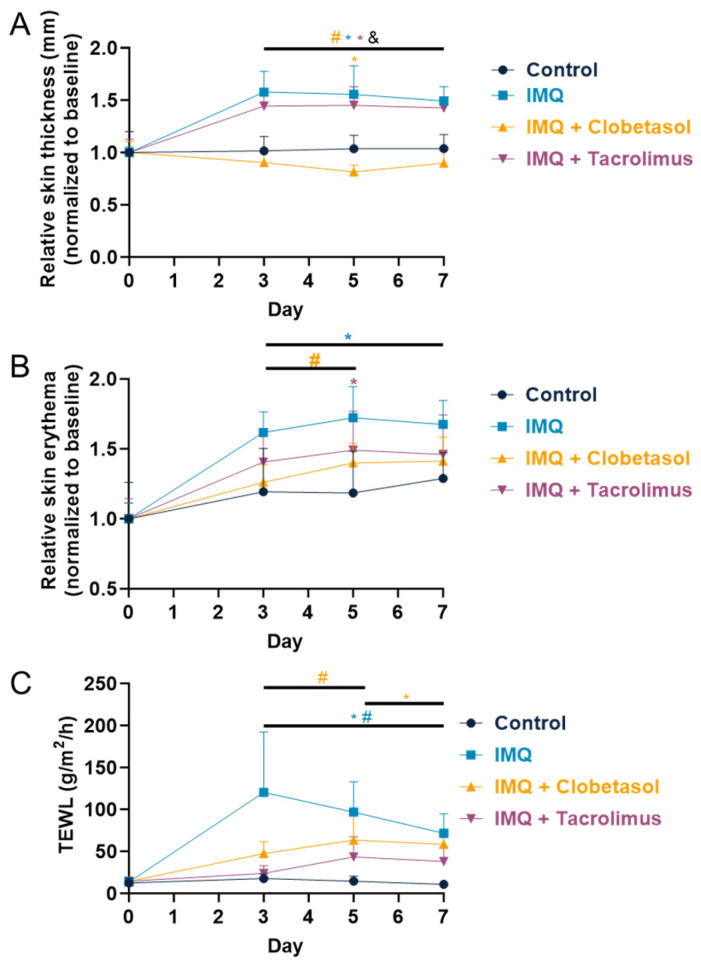
Effects of treatments on female rats’ skin inflammation upon Imiquimod: (**A**) Evaluation of the skin thickness in mm using a Mexameter upon Imiquimod (IMQ, topical administration, 37.5 mg/cm^2^) with or without Clobetasol (topical administration, 30 mg/cm^2^) or Tacrolimus (topical administration, 40 mg/cm^2^). Treatments were administered daily for up to 7 days. (**B**) Evaluation of skin erythema upon treatments. (**C**) Evaluation of Transepidermal Water Loss (TEWL) upon treatments. Statistical differences between the groups were determined using a mixed-effects model (REML, groups, and time as a factor) followed by Tukey’s multiple comparisons test: * *p* ≤ 0.05 control vs. treatments, # *p* ≤ 0.05 IMQ vs. IMQ + Clobetasol or IMQ + Tacrolimus, and & *p* ≤ 0.05 IMQ + Clobetasol vs. IMQ + Tacrolimus. Data represent mean and SD. n = 8 rats per group.

**Figure 4 ijms-25-09254-f004:**
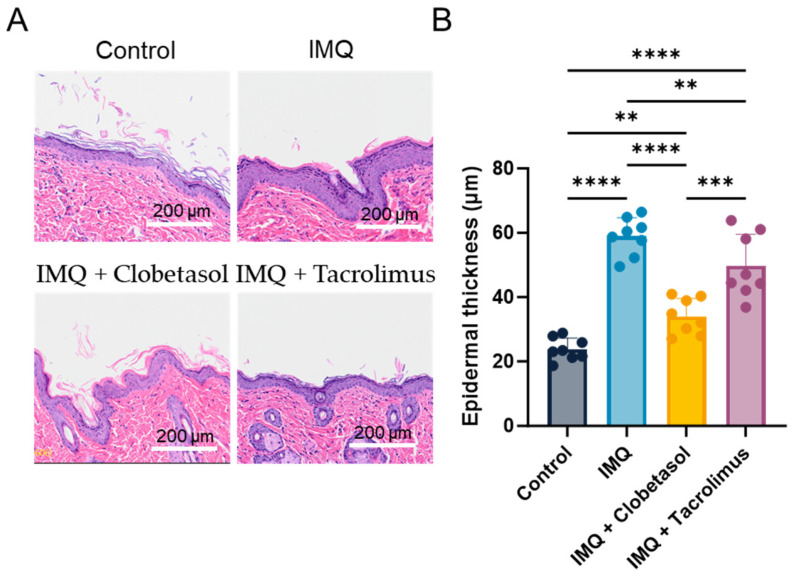
Characterization and evaluation of Imiquimod-induced psoriasis model in female rats: (**A**) Representative image of Hematoxylin and Eosin (HE) staining showing epidermal damage and thickness on Day 7 upon Imiquimod (IMQ, topical administration, 37.5 mg/cm^2^) with or without Clobetasol (topical administration, 30 mg/cm^2^) or Tacrolimus (topical administration, 40 mg/cm^2^). (**B**) Skin epidermal thickness quantification using image analysis. Data were expressed as mean ± SD, n = 8. Statistical differences between the groups were determined using a one-way ANOVA followed by Tukey’s multiple comparisons test: ** *p* ≤ 0.01, *** *p* ≤ 0.001, and **** *p* ≤ 0.0001. Scale bar: 200 µm.

**Figure 5 ijms-25-09254-f005:**
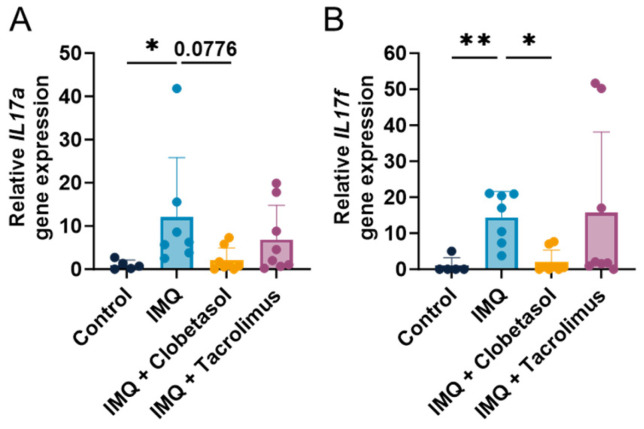
Effects of treatments on female rats’ Interleukin 17 gene expression upon Imiquimod: (**A**) Evaluation of the *IL17a* gene expression using RT-qPCR on Day 7 upon Imiquimod (IMQ, topical administration, 37.5 mg/cm^2^) with or without Clobetasol (topical administration, 30 mg/cm^2^) or Tacrolimus (topical administration, 40 mg/cm^2^). Treatments were administered daily for up to 7 days. (**B**) Evaluation of the *IL17f* gene expression using RT-qPCR on Day 7. Statistical differences between the groups were determined using a one-way ANOVA followed by Tukey’s multiple comparisons test: * *p* ≤ 0.05 and ** *p* ≤ 0.01. Data represent mean and SD. n = 8 rats per group.

**Figure 6 ijms-25-09254-f006:**
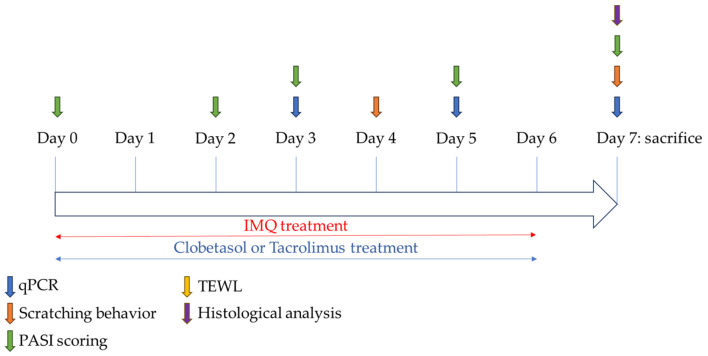
Experimental design of the Imiquimod IMQ-induced psoriasis in rats.

**Table 1 ijms-25-09254-t001:** Primer sequences used for RT-qPCR.

Target	Sequence (5′->3′)	Product Length
IL-17a	F: CCCTTAGCTCAAAAGCGAGC R: TCATGTGGTGGTCCAACTTCC P: CACAGTGCCGCCACCAGCGC	174
IL-17f	F: TCAATCAAAACCAGGGCATT R: TGATACAGCCTGAGTGTCTG P: ACCCGAGACCCCGACCGGTTCCCC	141
GAPDH	F: TTCAACGGCACAGTCAAG R: CCAGTAGACTCCACGACATA P: CCCATCACCATCTTCCAGGAGC	134

F, forward; R, reverse; IL, interleukin; GAPDH: glyceraldehyde-3-phosphate dehydrogenase.

## Data Availability

The data presented in this study are available upon request from the corresponding author. Any request should be addressed to Philippe Guillaume or Sonia Goineau at pguillaume@porsolt.com and sgoineau@porsolt.com, respectively.

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
