# Peer review of "Evaluation of Clobetasol and Tacrolimus Treatments in an Imiquimod-Induced Psoriasis Rat Model"

_ijms, 2024, doi:10.3390/ijms25179254_

Round 1
Reviewer 1 Report
Comments and Suggestions for Authors
First, I would like to congratulate the authors on their valuable work.
As the authors mentioned, psoriasis is one of the most frequent chronic inflammatory skin diseases. As no curative treatment exists, it continues to represent an appealing research topic. Identification of novel therapeutic targets and of methods to augment the efficacy of conventional therapies are equally important. The paper presents a concise, yet critical summary regarding the various psoriasis models, focusing on imiquimod-induced psoriasis in mouse models and provides a comprehensive analysis of the effects of two conventional topical treatments, clobetasol and tacrolimus, on the much less investigated rat model for psoriasis.
The manuscript is well organized, the study design is clear, and the results are presented in a straightforward manner. The use of figures and diagrams makes the data more accessible and understandable. The authors’ comments on their findings in light of previous studies represent a balanced, unbiased assessment. The conclusions are consistent with the evidence presented. The references are relevant and cover the most recent developments.
However, there are a few aspects that need to be addressed, as follows
· Editing of English language is required
· The phrase “Corticosteroids, such as Clobetasol and Tacrolimus, used as macrolide calcineurin 59 inhibitors, are widely used [4,6].” (lines 59 - 60) should be reformulated
· The phrase “Conversely, in male rats we did not observe a significant increase of scratching behavior following IMQ application, but scratching was significantly promoted by Clobetasol (Figures S1B).” (lines 107-109) contradicts the results depicted in Figure S1B and subsequent comments.
· A more detailed presentation of psoriasis pathogenesis should be included in the discussion section. (lines 236-238)
· “Difference male vs. female upon Clobetasol, similar effect on body weight, more scratching in male limited effect on PASI score.” (lines 306-307) should be reformulated to better illustrate the findings.
· “Although Tacrolimus has been shown to be an effective monotherapy for psoriasis, it does not always work well.” (lines 314-315) Please add references
· I suggest mentioning newer trends in psoriasis topical treatments that include induction and maintenance, proactive therapies.
· “Tacrolimus is known as a challenging molecule due to low solubility, low-penetration, poor-bioavailability, and toxicity.” Please add reference
Best regards!
Comments on the Quality of English LanguageEditing of English language is required
Author Response
Dear,
First of all, we wish to thank you for the constructive comments about our manuscript.
We have made responses to all comments as requested and we revised the manuscript accordingly. We resubmit the revised manuscript together with the point-to-point responses to the comments below. Thanks for the global positive feeback regarding the quality of our manuscript even if some adjustment and re-writing was requested.
Best regards,
Tristan Rupp, PhD
Comments from Reviewer
First, I would like to congratulate the authors on their valuable work.
As the authors mentioned, psoriasis is one of the most frequent chronic inflammatory skin diseases. As no curative treatment exists, it continues to represent an appealing research topic. Identification of novel therapeutic targets and of methods to augment the efficacy of conventional therapies are equally important. The paper presents a concise, yet critical summary regarding the various psoriasis models, focusing on imiquimod-induced psoriasis in mouse models and provides a comprehensive analysis of the effects of two conventional topical treatments, clobetasol and tacrolimus, on the much less investigated rat model for psoriasis.
The manuscript is well organized, the study design is clear, and the results are presented in a straightforward manner. The use of figures and diagrams makes the data more accessible and understandable. The authors’ comments on their findings in light of previous studies represent a balanced, unbiased assessment. The conclusions are consistent with the evidence presented. The references are relevant and cover the most recent developments.
Authors’ response: We would like to thank first the reviewer for the very constructive comments on our manuscript and we hope that the response and new edit of the manuscript will convince the reviewer of the interest of our work.
However, there are a few aspects that need to be addressed, as follows
- Editing of English language is required
Authors’ response: We edited the English of our manuscript thanks to a native speaker collaborator.
- The phrase “Corticosteroids, such as Clobetasol and Tacrolimus, used as macrolide calcineurin 59 inhibitors, are widely used [4,6].” (lines 59 - 60) should be reformulated
Authors’ response: Thanks for this comment, we reformulated this sentence for better clarity (lines 60-61).
- The phrase “Conversely, in male rats we did not observe a significant increase of scratching behavior following IMQ application, but scratching was significantly promoted by Clobetasol (Figures S1B).” (lines 107-109) contradicts the results depicted in Figure S1B and subsequent comments.
Authors’ response: Thanks for this comment, we reformulated this results description (lines 108-112). Indeed, we already observed in our previous finding that Clobetasol is able to reduce scratching in female rats as observed in this study (Lariosa-Willingham et al., BMC Res Notes. 2023 Nov 25;16(1):348). Globally, we observed a reduction of body weight, scratching behavior, and PASI score in IMQ-induced psoriasis female rats upon Clobetasol in our study. Consistently with the observation in female rats; we demonstrated in male rats a reduction of body weight and PASI score, however we did not observe a reduction of scratching behavior in male as observed in female rats. We think that this absence of expected effect might be due to hypersensitivity in male rats. Indeed, psoriasis induces a mild chronic stress due to the pathogenesis and associated disorders (Wang et al., J Leukoc Biol. 2020 Jul;108(1):267-281.). Interestingly, male rats are more sensitive to stress than female rats that can worsen the symptom including itch hypersensitivity (Takanami et al., Gen Comp Endocrinol. 2023 Aug 1:339:114289.). Thus, we think that despite a consistent response between males and females upon Clobetasol, the effect on scratching might be delayed and further analysis using longer treatment might be required to observe an effect on scratching behavior in male rats.
- A more detailed presentation of psoriasis pathogenesis should be included in the discussion section. (lines 236-238)
Authors’ response: Thanks for this comment, we edited accordingly (lines 249-260).
- “Difference male vs. female upon Clobetasol, similar effect on body weight, more scratching in male limited effect on PASI score.” (lines 306-307) should be reformulated to better illustrate the findings.
Authors’ response: Thanks for this comment, we reformulate this data description including the above description (line 328-339).
- “Although Tacrolimus has been shown to be an effective monotherapy for psoriasis, it does not always work well.” (lines 314-315) Please add references
Authors’ response: Thanks for this comment, we reformulate to attenuate the statement and we cited the article of Malecic and Young (Psoriasis (Auckl). 2016; 6: 153–163.) that described of the limitation of use for Tacrolimus (lines 346-347).
- I suggest mentioning newer trends in psoriasis topical treatments that include induction and maintenance, proactive therapies.
Authors’ response: We thank the reviewer for this interesting comment, and we edited accordingly (line 360-368).
- “Tacrolimus is known as a challenging molecule due to low solubility, low-penetration, poor-bioavailability, and toxicity.” Please add reference
Authors’ response: Thanks for this comment, we edited accordingly by referring to the article of Patel and collaborators (Int J Pharm Investig. 2012 Oct-Dec; 2(4): 169–175) (line 355).
Best regards!
Reviewer 2 Report
Comments and Suggestions for Authors
This article established a psoriasis-like model in rats induced by IMQ and investigated the therapeutic effects of Clobetasol and Tacrolimus on psoriatic lesions. The overall structure of the manuscript is well-established, while several issues need to be addressed:
-
Expression: The English is generally accurate but contains several problems highlighted in the PDF. Instead of using parentheses for supplementary explanations, please use complete sentences to enhance clarity and readability.
-
Method: The method section lacks clarity regarding the timeline for animal modeling and sample collection. It is recommended to revise this section or add a timeline figure like Reference [9].
-
Results: The content of the animal experiments is relatively limited, especially concerning the basic mechanisms. For rats, only HE staining and qPCR detection of IL-17a/f gene expression were used. If possible, consider adding immunohistochemistry, Western blot, or ELISA experiments to further verify the expression of IL-17a/f and even downstream proteins in the IL-17 pathway. This would make the exploration of the basic mechanisms more comprehensive.
This study offers valuable data on drug treatment in the rat model of psoriasis. While the article's innovation and scope are currently somewhat limited, expanding the study's depth or workload could improve its overall quality, and overall significance.

The English is generally accurate but contains several problems highlighted in the PDF. Instead of using parentheses for supplementary explanations, please use complete sentences to enhance clarity and readability.
Author Response
Dear,
First of all, we wish to thank you for the constructive comments about our manuscript.
We have made responses to all comments as requested and we revised the manuscript accordingly. We resubmit the revised manuscript together with the point-to-point responses to the comments below. Thanks for the global positive feedback regarding the quality of our manuscript even if some adjustment and re-writing was requested.
Best regards,
Tristan Rupp, PhD
Comments from Reviewer
This article established a psoriasis-like model in rats induced by IMQ and investigated the therapeutic effects of Clobetasol and Tacrolimus on psoriatic lesions. The overall structure of the manuscript is well-established, while several issues need to be addressed:
- Expression: The English is generally accurate but contains several problems highlighted in the PDF. Instead of using parentheses for supplementary explanations, please use complete sentences to enhance clarity and readability.
Authors’ response: Thanks for this comment, we edited the English of our manuscript thanks to a native speaker collaborator including the comments from the reviewer.
- Method: The method section lacks clarity regarding the timeline for animal modeling and sample collection. It is recommended to revise this section or add a timeline figure like Reference [9].
Authors’ response: We would like to thank first the reviewer for the constructive comment. Indeed, we also think that a timeline figure enhances the clarity of the work (lines 423-425 and Figure 6).
- Results: The content of the animal experiments is relatively limited, especially concerning the basic mechanisms. For rats, only HE staining and qPCR detection of IL-17a/f gene expression were used. If possible, consider adding immunohistochemistry, Western blot, or ELISA experiments to further verify the expression of IL-17a/f and even downstream proteins in the IL-17 pathway. This would make the exploration of the basic mechanisms more comprehensive.
Authors’ response: We understand the request from the reviewer. In a previous study, we evaluated the expression of IL-17a at protein level and demonstrated that IL-17a is not detectable in blood samples from rat treated with IMQ contrary to human blood samples from psoriatic patient can be expressed in situ into skin samples treated with IMQ (Lariosa-Willingham et al., BMC Res Notes. 2023 Nov 25;16(1):348 / Bai et al., Oncotarget. 2018;9(1):1266–78.). IL-17a is also poorly detectable in rat skin but is detectable in ear skin treated with IMQ (Lariosa-Willingham et al., BMC Res Notes. 2023 Nov 25;16(1):348). In our study we thus favored the sampling of skin samples for pure transcriptomic to potentiate the detection of IL-17 gene expression and evaluate the effect of Clobetasol. Interestingly, several articles demonstrated that IL-17 is overexpressed in psoriasis condition in mouse and human skin samples (Harper et al., J Invest Dermatol. 2009 Sep;129(9):2175-83. / Aramwit et al., Sci Rep. 2023 Jul 26;13(1):12133. / Qin et al., Mol Med Rep. 2014 Jun;9(6):2097-104.). Moreover, Clobetasol has shown to reduce IL-17 protein expression in mouse model of psoriasis (Almudaris and Gatea. Pharmacia 2024 Feb;71:1-14.). Altogether, these observations confirmed consistently with our data that IL-17 expression is upregulated by IMQ and repressed by Clobetasol.
This study offers valuable data on drug treatment in the rat model of psoriasis. While the article's innovation and scope are currently somewhat limited, expanding the study's depth or workload could improve its overall quality, and overall significance.
Authors’ response: We would like to thank first the reviewer for the very constructive comments on our manuscript and we hope that the response and new edit of the manuscript will convince the reviewer of the interest of our work.
Round 2
Reviewer 2 Report
Comments and Suggestions for Authors
The issues I raised during the last revision of this manusript have been well addressed by the authors.